



# On the Additivity of Climate Responses to the Volcanic and Solar Forcing in the Early 19th Century

Shih-Wei Fang[1], Claudia Timmreck[1], Johann Jungclaus[1], Kirstin Krüger[2], Hauke Schmidt[1]

[1]Max-Planck-Institut für Meteorologie, Hamburg, 20146, Germany

[2]Department of Geosciences, University of Oslo, Oslo, 0371, Norway

*Correspondence to*: Shih-Wei Fang (shih-wei.fang@mpimet.mpg.de)

**Abstract.** The early 19th century was the coldest period over the past 500 years, when strong tropical volcanic events and a solar minimum coincided. The 1809 unidentified eruption and the 1815 Tambora eruption happened consecutively during the Dalton minimum of solar irradiance; however, the relative role of the two forcing (volcano and solar) agents is still unclear. In

this study, we examine the effects from combinations of one volcanic with two different solar forcing reconstructions (SATIRE and PMOD) suggested in the protocol for the past1000 experiment of the Paleoclimate Modelling Intercomparison Project - Phase 4 (PMIP4) by simulating the early 19th century climate. From 20-member ensemble simulations with the Max Planck Institute Earth System Model (MPI-ESM1.2-LR), we find that the volcano- and solar-induced surface cooling is in general additive, regardless of combining or separating the forcing agents. The two solar reconstructions (SATIRE and PMOD)

contribute on average ~0.05 K/month and ~0.15 K/month surface air cooling, respectively, indicating a limited solar contribution to the early 19th century cold period. The volcanic events provide the main cooling contributions, inducing a surface cooling peak of ~0.82 K for the 1809 event and ~1.35 K for Tambora. After the Tambora eruption, the cooling in most regions reduces largely within 5 years when a global cooling of ~0.34 K is reached, along with the reduction of volcanic forcing. In the northern extratropical oceans, the cooling reduces only slowly with a constant rate until 1830, which is related

to the reduction of seasonality and the increased Arctic sea-ice extent. Also, the albedo feedback of Arctic sea ice is found to be the main contributor to the Arctic amplification of the cooling signal. Several non-additive responses to solar and volcanic forcing happen on regional scales. In the atmosphere, the polar vortex tends to strengthen when combining both volcano and solar forcing, even though the two forcing agents separately induce opposite responses. In the ocean, when combining the two forcings, additional surface cold water propagates to the northern extra-tropics from the additional solar cooling in the tropics,

which results in regional cooling along the propagation. Overall, this study not only quantifies the surface responses from combinations of volcano and solar forcing, but also highlights the components that cannot be simply added from the responses of the individual forcing agents, indicating that a relatively small forcing agent (such as solar in early 19th century) can impact the response from the large forcing (such as the Tambora eruption) when considering regional climates.



## 1 Introduction

Being the coldest period over the past 500 years, the early 19th century is a crucial period for studying the climate impacts from natural external forcing such as volcanoes and solar irradiance (Cole-Dai et al., 2009; Brönnimann et al., 2019). With limited impacts from anthropogenic greenhouse gas, the on-average low temperature in the early 19th century is believed to be caused mainly by the coincidental existence of strong tropical eruptions (the unidentified 1809 and the 1815 Tambora eruptions; Self et al. 2004; Cole-Dai et al., 2009) and the lower solar irradiance (Dalton minimum from 1790-1830; Usoskin

et al. 2002; Silverman and Hayakawa, 2021). Studies have investigated the climate impacts from the 1809 unidentified (Timmreck et al., 2021), the 1815 Tambora eruptions (Raible et al., 2016; Schurer et al., 2019; Zanchettin et al., 2019), and the Dalton minimum (Anet et al., 2014). For example, Zanchettin et al. (2019) examined how different strengths of the unidentified 1809 eruption could have altered the cooling caused by the following 1815 Tambora eruption. However, studies on the interplay between the solar- and volcanic-induced climate responses in the early 19th century are rare. Wagner and

Zorita (2005) found that volcanoes are the main contribution to the winter surface cooling in the early 19th century, but no winter cooling is found with the solar-only simulations. On the other hand, Anet et al. (2014) argued that the volcanic forcing alone cannot reproduce the long-lasting cooling found in the reconstructions and the solar forcing is essential for the post-volcanic cooling. Both modeling studies considered only three ensemble members, resulting in not only the lack of quantification of the individual contributions from volcanic and solar forcing but also an inability to identify potential non-

additive responses when combining both forcing.

For the quantification of the solar and volcanic forcing in the early 19th century, the uncertainty of the reconstructions used for forcing agents is a challenge. For instance, reconstructions of total solar irradiance (TSI) and spectral solar irradiance (SSI) can be calculated by various proxies, such as the sunspot number, solar modulation potential, and cosmogenic isotope concentrations of $^{14}C$ and $^{10}Be$ from tree rings and ice cores (Usoskin, 2017). Depending on the method, the estimated

ultraviolet irradiance can differ by up to 15% (Shapiro et al., 2011). As a result, two solar reconstructions and one volcano forcing are recommended in the protocol of the experiments covering the Common Era (Jungclaus et al., 2017) in the Paleoclimate Modelling Intercomparison Project - Phase 4 (PMIP4; Kageyama et al., 2018). This protocol provides a more consistent way for quantifying the early 19th century climates and further examining the interplay between the volcanic and solar forcing. The two solar reconstructions, SATIRE (the updated Spectral and Total Irradiance REconstruction-Millennia

model; Vieira et al., 2011; Wu et al., 2018) and PMOD (Physikalisch-Meteorologisches Observatorium Davos; Shapiro et al., 2011; Egorova et al., 2018) may allow estimating possible boundaries of how much the solar forcing may contribute to the past climates as the PMOD has a larger variability and can be considered as an upper limit for the reduction of solar radiation (Schmidt et al., 2012; Jungclaus et al., 2017).

The interplay of volcano- and solar-induced impacts may not be straightforward since both volcanic aerosol and lower solar

irradiance can induce surface temperature changes along two pathways: bottom-up and top-down. The bottom-up processes caused by the two forcing agents cool the surface temperature with the reduction of incoming solar radiation at the surface due



to the lower solar irradiance (Meehl et al., 2009; Misios and Schmidt, 2012) and the blocking of incoming solar radiation from the volcanic aerosols (Robock, 2000; Timmreck, 2012; Swingedouw et al., 2017). In contrast, the top-down processes for the two forcings are opposite: the reduced solar radiation can cool the lower stratosphere, reduce the meridional temperature
gradient, and weaken the polar vortex (Scaife et al., 2013; Maycock et al., 2015), while the volcanic aerosols can warm the lower tropical stratosphere causing a strengthening of the polar vortex (Robock, 2000; Timmreck, 2012; Swingedouw et al., 2017). These opposite polar vortex responses may result in distinct surface responses, especially at high-latitudes. A question is, when both solar and volcanic forcing are included in the system, can the top-down and bottom-up processes of both forcings be simply additive? And how will the surface temperature cooling respond to the combination of both forcings and both
processes?

Besides, Arctic Amplification (AA) is a possible factor creating the latitudinal difference of surface air cooling on Earth as AA is well-known for the latitudinal difference of warming (Graversen et al., 2008; Serreze et al., 2009; Previdi et al., 2021). Under global warming, the Arctic region is observed and expected to warm faster than other regions and to reduce the seasonality due to a stronger AA in winter. In fact, AA may also exist when the system undergoes a surface cooling. For
example, Stjern et al. (2019) show that solar irradiance change has a similar AA strength as other drivers (such as $CO_2$). Liu et al. (2018) state that no volcano-induced AA is found due to the large temperature changes in the tropics, while others have found that the northern extra-tropics encounter a larger cooling a few years after eruptions (Zanchettin et al., 2019). This means that the extent of AA for volcanic and solar forcing is not well understood and the combination of both has not yet been studied. The paper is structured as follows: Section 2 discusses the data, method, and experiment design; Section 3 describes the impacts
from solar and volcano forcing separately; Section 4 investigates the additivity between the volcanic and solar signals at the surface, atmosphere, and ocean; Section 5 studies the AA during the solar- and volcanic-forced cooling with a feedback separation; and Section 6 summarizes the paper and discusses potential follow-up studies.

## 2 Datasets and methods

In this study, the Max Planck Institute for Meteorology Earth System Model version 1.2 at low resolution (MPI-ESM1.2-LR),
which is the CMIP6/PMIP4 version of MPI-ESM (Mauritsen et al., 2019), is used for our simulations. The MPI-ESM1.2-LR is a state-of-art climate model composed by the ECHAM6.3 atmospheric model, JSBACH3.2 land model, the MPIOM1.6 ocean model, the HAMOCC6 ocean biogeochemistry model, and the OASIS3-MCT coupler. The atmospheric component is using the T63 Atmospheric triangular truncation (~200 km horizontal resolution) with 47 vertical levels and the ocean grid is the GR1.5 grid (~150 km nominal horizontal resolution) with 40 vertical levels. For solar forcing, the SSI is interpolated to 14
wavelength bands with the lowest bound at 120 nm. The stratospheric aerosol forcing is prescribed with monthly zonal mean aerosol optical properties and without interactive chemistry. It is noted that the quasi-biennial oscillation (QBO) is not simulated in this version of MPI-ESM due to the relatively low vertical resolution (Krismer et al., 2013). This may impact our results as the QBO is known to affect the Arctic oscillation (Holten and Tan 1980; Garfinkel et al. 2012; Labe et al., 2019).



For instance, Stenchikov et al. (2004) show that the easterly phase of QBO right after the 1991 Pinatubo eruption tends to

weaken the polar vortex, while the westerly phase of QBO in 1992-1993 can enhance the aerosol effect on the AO.

Four sets of ensemble experiments, each consisting of 20 members, are conducted in this study (Table 1). All experiments are carried out from the year 1791 to year 1830 for 40 years, where the 20 ensemble members are created by perturbing the atmosphere at the beginning of 1771 from the MPI-ESM past2k run (van Dijk et al., 2021; Fang et al., 2021) and simulations are run from 1771-1790 to obtain distinct ocean states (Fig. 1a). Besides the volcanic aerosol and solar activity (including the

ozone concentration in the upper atmosphere), all other boundary conditions of the experiments are the same as in the past2k simulation.

The first two experiments, Basic and SolarStrong, are simulated with the SATIRE-14C (Usoskin et al., 2016) and PMOD-14C (Shapiro et al., 2011; Egorova et al., 2018) solar reconstructions formulated in the PMIP4 past1000 protocol. The reason to use two solar reconstructions is because no prior research, to our knowledge, has quantified the possible range of solar impacts

in the early 19[th] centuries with two different reconstructions. The PMOD reconstruction has a larger variability with lower solar radiation in the early 19[th] century than the SATIRE reconstruction (Fig. 1b) due to their distinct assumptions of the proxy, the solar modulation potential and the photospheric magnetic field changes, respectively. This entails why the PMIP4 protocol provides two reconstructions since the PMOD reconstruction can be considered as an upper limit of the possible secular variability as it generally shows a larger variability than other reconstructions (Schmidt et al., 2012; Jungclaus et al., 2017).

Specifically, we use the 14C version of the reconstructions calculated from the cosmogenic isotope concentrations of 14C. The prescribed ozone concentration is calculated based on solar irradiance (not including the volcanic impacts) following the PMIP past1000 protocol (Jungclaus et al., 2017; Matthes et al., 2017), as the solar irradiance can alter the stratospheric and mesospheric ozone concentration (Haigh, 1994). The solar effects on ozone for the past1000 experiment are scaled using the averaged solar UV irradiance (from 200 to 320 nm) because the 10.7 cm radio flux (F10.7) used for the CMIP6 historical

simulation is not available for the past1000 time period.

The other two experiments are Volcano (volcanic aerosol + SATIRE) and Volcano&SolarStrong (volcanic aerosol + PMOD) experiments, which include the volcanic aerosols but different solar reconstructions to disentangle the interaction between the solar and volcanic cooling in the early 19[th] century. Following the PMIP4 past1000 protocol, the time varying sulfur injections of volcanic forcing is calculated with the EVA (Easy Volcanic Aerosol) module (Toohey et al., 2016) using the eVolv2k

dataset (Toohey and Sigl, 2017) as input. Two strong volcanic eruptions occur during the experiment period: the 1809 and the 1815 Tambora eruptions (Fig. 1c). With the combinations of with/without volcanic aerosol and strong/weak solar irradiance we aim to better understand the possible contributions of different forcings to the early 19th cooling.

The anomalies in this study are calculated based on the ensemble mean of the seasonal cycle of the Basic experiments (anomalies = each ensembles minus the ensemble mean climatology of Basic). This is because the Basic experiment has the

smallest forcing of surface cooling (SATIRE-14C and no volcanic aerosol). Several climate indices are used in this study. The Niño3.4 index is defined as the regional mean sea surface temperature (SST) anomalies over 5° S to 5° N and 170° W to 120° W. The relative Niño3.4 index, which represents the ENSO responses without impacts from global volcanic cooling, follows





Khodri et al. (2017) by removing the tropical mean (20° S to 20° N) SST anomalies from the Niño3.4 index. The North Atlantic Oscillation (NAO) index (Wanner et al., 2001) is calculated by the difference of sea level pressure anomalies between the
Azores (36° N to 40° N and 28° W to 20° W) and the Icelandic low (63° N to 70° N and 25° W to 16° W). The Arctic Oscillation (AO; Thompson and Wallace 1998) index is obtained by the normalized first principal component (PC) of the Empirical Orthogonal Function (EOF) of northern extratropical sea level pressure anomalies (20° N to 90° N). The Southern Annular Mode (SAM; Marshall 2003) index is computed by the zonal mean sea level pressure anomalies between 40° S and 65° S. The Pacific Decadal Oscillation (PDO; Mantua and Hare 2002) is obtained by the normalized first PC of the northern
extratropical Pacific SST anomalies (20° N to 50° N and 120° E to 110° W) EOF. The Atlantic multi-decadal variability (AMV; Enfield et al., 2001) is the 10-year running mean of the north Atlantic SST anomalies (0° N to 60° N and 80° W to 0° W) and the Atlantic meridional overturning circulation (AMOC) is defined as the maximum Atlantic meridional overturning streamfunction at 30° N below 1000 m depth. For the significance of each month of indices for an experiment, the 20 ensembles of an experiment is compared with the 20 ensemble of the Basic experiment by the student-t test.
To investigate what process contributes the most to AA cooling, we use the rapid adjustment (Marshall et al. 2020) that disentangle the radiative contributions of each feedback in the Arctic and Tropics. The rapid adjustment is computed with two steps: 1) obtains the radiative kernel (Block and Mauritsen 2013) from the piControl run simulation of the same model (MPI-ESM1.2-LR), which we can get how much radiation will change when the climate deviates (e.g., how much Planck feedback will change when surface air temperature change); 2) multiplies the kernel with the difference between the experiments and
piControl run of corresponding variables for the feedback to estimate the radiative changes of each process (e.g., multiplies the kernel for Planck feedback with the surface air temperature difference between the experiments and piCotronl). The reason we used the rapid adjustment is that our forcing agents (solar and volcano) change over time and we are considering a shorter period compared to general methods of radiative kernels (decades).

## 3 The Surface Impacts from Solar and Volcano Forcing

Figures 2a and 2e show the global mean surface air temperature (SAT) and SST of the Basic and SolarStrong experiments. As a baseline for the other experiments, the Basic experiment (blue) has small temperature changes over this period, where the temperature bounces back to normal between 1795-1800 from the Icelandic Laki eruption in 1783, which is inherited from the initial conditions (Fig. 1a). Also, the weaker (stronger) solar radiation before (after) 1815 causes a slightly lower (higher) temperature, meaning the SATIRE solar forcing during the Dalton minimum results in less than 0.03 K/month global SST
cooling (< 0.05 K/month for SAT) in the MPI-ESM1.2-LR. The spatial averages of different regions (Figs. 2b-d) also show similar temperature variations, but the tropical region (30° S to 30° N) exhibits a weaker temperature change compared to the southern extratropical (90° S to 30° S) and the northern extra-tropics (30° N to 90° N).

The SolarStrong experiment (orange) includes lower incoming solar radiation than the Basic experiment after 1795. Their difference increases after 1795, peaks around 1815 and reduces afterward (Fig. 1b). Compared to Basic, the global mean SSTs





in SolarStrong are significantly different after 1805 with a ~0.07 K SST (~0.1 K SAT) difference throughout the period (Figs. 2a and 2e). The significant cooling (student-t tests between the 20 ensemble-members of Basic and SolarStrong; shown in the orange rectangle below the time series) is mainly found in the northern extra-tropics (Figs. 2c and 2g) while the tropical (Figs. 2b and 2f) and southern extratropical (Figs. 2d and 2h) regions have limited differences. Maximum differences in solar irradiance and surface temperature do not coincide. The solar irradiance shows the strongest difference in 1815, while the

strongest temperature differences occur around 1812 and 1820, when the tropics and southern extra-tropics (90°S to 30°S) exhibit significant differences compared to Basic. This indicates that the solar forcing does not exhibit clear instantaneous responses and the small cooling responses can easily be hidden by the internal variability if small ensembles are considered. Figures 3a and 3e show the spatial structure of the mean SAT anomalies in winter and summer from SolarStrong over 1809-1826. This period will be used for the rest of the analyses since it covers the largest solar irradiance difference (~1815), the

1809 and 1815 eruptions, and the post-eruption responses (1821-1826). The largest temperature difference to Basic is found in the Arctic region in Winter as a result of AA. Significant differences for individual grid points are mainly simulated in the tropical region but rarely in the extra-tropics. This indicates the difficulty of detecting surface cooling responses from the reduced-solar forcing in specific regions. Even for the northern extratropical cooling that is significant in the spatial average (Fig. 2c), the cooling is not persistently located in specific regions but varies between the realizations. In fact, significant

regions in SST and SAT are often collocated (Figs. 3b and f), indicating the control of SST on SAT in these regions. For the SST, the northern hemisphere shows mostly negative values except in the Gulf stream and Kuroshio regions, echoing the stronger cooling signatures in the northern extra-tropics. The positive anomalies in the Gulf Stream and Kuroshio and the strong cooling in the north-east of the Pacific and Atlantic basins manifest the weakening of the wind-driven circulation, which may be due to the weaker westerly winds (Kwon and Joyce, 2013) and the freshwater flux from Arctic sea ice changes

(Zanchettin et al., 2012). The weaker westerlies are associated with a negative phase of the AO (Fig. 3h) in boreal winter, but not in summer (Fig. 3d), even though the SST cooling patterns show not much difference. The increase of the Arctic sea-ice extent is found in all seasons (Figs. 3c and g), accompanied with a smaller increase in the summer than in winter. We interpret this stronger seasonality of Arctic sea-ice extent under solar forcing to be related to the smaller SST seasonality, where less heat is stored in the summer ocean, leading to the stronger sea-ice formation in winter (Carton et al., 2015). The reduction of

the westerlies may also be caused by the weakening of the polar vortex due to the smaller meridional temperature gradient in the lower stratosphere, which we will discuss in Section 4.2. In contrast, the Southern Ocean is characterized by a mix of warming and cooling signatures, which correspond to the regions of decrease and increase of Antarctic sea-ice extent, respectively. This entails why not much surface cooling is found in the southern extra-tropics.

The Volcano experiment (green lines; Figs. 4a and 4e) shows clear surface temperature decreases immediately after the 1809

and 1815 eruptions with a maximum of ensemble-mean SST cooling around -0.53 K and -0.94 K (-0.82 K and -1.36 K for SAT), respectively. As indicated in Figure 1c with the outgoing solar radiation, the effect of the prescribed volcanic~~aerosols~~ radiative forcing without interactive aerosols is mostly removed after 3-4 years; however, the temperature in all regions is not only significantly different right after the eruptions compared to Basic after 1809, but the negative anomalies continue even





when the direct volcanic effect is strongly reduced 5-6 years after the eruption. The tropical region (Figs. 4b and 4f) mainly
shapes the global cooling signature in the first 3-5 years, when volcanic forcing exists, while the northern extra-tropics (Figs.
4c and 4g) contribute to the cooling afterwards, and weaker (larger) seasonal variations are observed in the SST (SAT). Directly
responding to the aerosol forcing, the tropical cooling has a sharp increase followed by a decrease in the first few years after
the eruptions, and the SST cooling maintains at ~0.2 K afterwards. This is because the cooling is stored in the upper ocean and
slowly diffuses and is transported to the entire ocean.

For the northern extra-tropics (Figs. 4c and g), SAT and SST show a distinct response. After the direct responses to volcanic
eruptions, the SST cooling slowly reduces by ~0.1 K/year, while the SAT returns to around -0.5 K rapidly with the same pace
as other regions along with the diminishing volcanic forcing (even clearer if considering land-only). We interpret this slower
retrieval of SST to be related to interactions with the sea-ice extent as mentioned in solar-induced cooling (Carton et al. 2015),
where the SST cooling is stored in the mixed layer during summer to impede the summer sea ice loss and enhance the winter
sea ice formation. The reduced SST seasonality with the colder summer and the steady retrieval of the cooling is not found in
the southern hemisphere, as the Antarctic sea ice shows both increases and decreases depending on the region.

The cooling pattern of the Volcano experiment is similar to the SolarStrong experiment in general (Fig. 5), where the AA
results in larger SAT cooling in the high latitude in winter. The sea ice-related regions, such as the Gulf of Alaska and the
Labrador Sea, exhibit the most apparent SST cooling. Differences can be found in the sea-ice extent and sea level pressure.

The Volcano experiment shows more sea-ice extending to lower latitudes in both summer and winter due to the stronger
cooling from volcanic forcing. A positive AO (i.e., low pressure in the Arctic and high pressure in the midlatitude) is found in
boreal winter in the Volcano experiment, while SolarStrong has a negative AO with high pressure in the Arctic (Fig.3). This
different AO response is related to the distinct stratospheric responses from volcanic aerosol forcing and reduced solar
irradiance, which will be further discussed in Section 4.2. That is, although different patterns are found in the sea-ice extent
and surface pressure and the cooling magnitudes differ, the response patterns of surface temperature are similar in the Volcano
and SolarStrong experiments, indicating that the top-down mechanism via the AO has limited impacts and the surface is more
dominated with the bottom-up direct radiative cooling.

## 4 The Additivity of Volcanic and Solar Forcing

### 4.1 The Interplay at the Surface

The Volcano&SolarStrong experiment is conducted to analyze the combined response of the stronger solar forcing and the
volcanic aerosols. We find that the global average surface temperature response is, in general, additive when separately or
together imposing the volcanic and solar forcing, even for the seasonality changes in the northern extra-tropics and the slow
return of the northern extra-tropical SST. This additivity is clearly illustrated in Figures 6a-h by the similar values of the added
SolarStrong and Volcano signals (purple dashed lines) in comparison to the Volcano&SolarStrong experiment (red solid lines)
after the eruptions. The SST changes match even 7 years after the 1815 Tambora eruption with the Volcano&SolarStrong





experiment. To further study whether the additivity also exists spatially, Figure 7 shows the zonal mean Hovmöller plot of the winter-surface temperature difference of solar signals (Volcano&SolarStrong minus Volcano and SolarStrong minus Basic). The SAT responses are similar (Figs. 7c and 7f). The AA dominates the cooling response when the reduced solar radiation is imposed on both cases with/without volcanic forcing, while the SST has a distinct response. The reduced solar radiation in the case without volcanic forcing causes the largest SST cooling steadily remaining at around 50° N (Fig. 7i) and over time echos the large cooling at the Gulf of Alaska and the Labrador Sea in the SolarStrong experiment. On the other hand, when imposing the solar forcing in the case with volcanic forcing, a large cooling signature for the northern hemisphere is found mainly south of 50° N (Fig. 7l), with additional cooling related to the two eruptions. Around 1815, additional cooling exists in the western tropical Pacific and Atlantic (Fig. 8d), and then propagates to the higher latitudes through the ocean circulation of the western boundary currents (Fig. 8d-f). Large cooling can also be found in the Arctic (Norwegian Sea), related to the expansion of sea ice with reduced solar irradiance. These differences in spatial SST responses indicate that the regional response is different when solar forcing interacts with the volcanic forcing even though the large-scale average responses are additive.

## 4.2 The Interplay in the Atmosphere

Here, we further investigate the atmospheric responses to the volcanic and solar forcings by inspecting the zonal means of air temperature and zonal wind anomalies (Fig. 9) for the volcano (1809-1820) and the post-volcanio (1821-1826) periods. In the SolarStrong experiment (Fig. 9a) over the volcano period, the reduced solar irradiance results in a weaker polar vortex (shading) in the winter hemisphere due to the weaker meridional temperature gradient (contour). The westerly anomalies in the subpolar region can be found throughout the troposphere and stratosphere and reduce towards the surface to almost zero. That is, the reduced solar radiation can cool the high-latitude atmosphere through the top-down mechanism (though not significantly), but a limited response is found at the surface with no tendency for a negative AO index (Fig. 10d) even though a weak AO pattern can be observed in the sea level pressure (Fig. 3h). For the Volcano experiment, the direct responses from volcanic aerosols can be found in 1809-1820 (Fig. 9b). The warming in the tropical lower stratosphere enhances the meridional temperature gradient, leads to westerly anomalies around 25° N to 50° N, strengthens the polar vortex, and cools the polar stratosphere in the winter hemisphere. From the Volcano&SolarStrong experiment (Fig. 9c), we can see that the impacts from volcanic aerosol forcing overwhelms the cooling induced by the reduced solar irradiance plus ozone in this period due to the relatively larger magnitude of the volcanic forcing. To further investigate whether the responses are additive, Figure 9d shows the difference between the Volcano&SolarStrong and the addition of the SolarStrong and Volcano experiments. The additional westerly anomalies in the middle to upper stratosphere illustrate an even stronger polar vortex in the Volcano&SolarStrong experiment accompanied with a cooler polar stratosphere. This entails that the reduced solar radiation may contribute to a stronger polar vortex in the middle stratosphere when strong volcanic eruptions occur even though the temperature and zonal wind responses are in opposite directions to those from reduced solar radiation alone.





For the post-volcano period (1821-1826), the direct impact from the volcanic aerosol becomes weaker and the responses is comparable with the solar forcing. In the tropical lower stratosphere and upper troposphere, the temperature responses in Volcano&SolarStrong (Figs. 9g) show negative values, which is similar to the combination of the SolarStrong (Figs. 9e) and Volcano experiments (Figs. 9f). This may explain why no apparent response is found in the polar vortex and polar air temperature in the Volcano&SolarStrong experiment since the tropical stratospheric temperature changes from solar and volcanic forcing balance each other. In fact, if we inspect their difference (Fig. 9h), similar strengthening of the polar vortex and colder air temperature in the middle stratosphere is observed (though not significant) as in the previous period (Fig. 9d). This additional strengthening of the polar vortex when combining solar and volcanic forcing needs to be further examined. In addition, the signal is significant in the polar troposphere. This is related to a corresponding response in the same region in the Volcano experiment. This indicates that the volcanic impact is more related to the surface, while the responses to the reduced solar radiation are less significant and may be more confined to the stratosphere. To be noticed, the MPI-ESM1.2-LR used in this study has a prescribed ozone (not interacting with volcanic aerosol changes) and no QBO is simulated, which may impact our results, especially in the stratosphere (Anet et al. 2013).

### 4.3 The Interplay in Ocean and Climate Indices

The subsurface ocean responses are mostly additive (not shown) as they are related to the surface ocean changes, which are also additive for solar and volcanic forcing. Despite the additivity in most regions, the El Niño-Southern Oscillation (ENSO) shows a distinct response when combining the solar- and volcano-induced responses. The SolarStrong experiment has no apparent tendency of the Niño3.4 index, while the Volcano and Volcano&SolarStrong experiments have negative values after volcanic eruptions (Fig. 10a). These negative values are, in fact, not a La Niña response but a global cooling signature from the eruptions. The actual ENSO activities can be estimated by the relative Niño3.4 index (Fig. 10b). By removing the tropical mean SST, an El Niño response after an eruption is often found in model simulations (Khodri et al., 2017). In the Volcano experiment, the 1809 eruption is followed by strong El Niño signatures for two consecutive years, while no apparent ENSO activity is found after the 1815 Tambora. This is because there is an El Niño signature in 1814/15 that tends to trigger a La Niña in the 1815 winter and suppress the El Niño tendency after Tambora. On the other hand, the Volcano&SolarStrong experiment has weak El Niño signatures for two years after both 1809 and 1815 eruptions, indicating that the reduced solar irradiance may cause a smaller El Niño tendency after the 1809 eruption. This complex ENSO response to volcanic and solar forcing needs to be studied in detail in the future.

We further investigate the solar- and volcano-induced responses for other climate indices. The higher latitudes are characterized by the NAO, AO, and SAM. Positive NAO (Fig. 10c) and AO (Fig. 10d) phases are simulated right after the volcanic eruption, signals which are often also observed in the first winter after eruptions (Christiansen, 2008). No significant signal is simulated for the 1810 winter and not all the 1816 winter has a significant signal, which shows that the large internal variability dominates in the northern extratropical region. After the first year, when the northern extra-tropics also encounter stronger cooling, no significant impact is found for NAO and AO, which agrees with the weak response in the composite of





sea level pressure (Figs 3h and 5h), and indicates the limited impact from the top-down mechanism in the MPI-ESM-LR1.2. For the southern hemisphere, no apparent signal is found in SAM (Fig. 10e), which is consistent with the mix of increasing and decreasing Antarctic sea-ice related to the surface pressure and temperature changes in the same regions.

Multi-decadal climate responses can be observed for the PDO, AMV, and AMOC. The PDO does not respond significantly to
the solar forcing, while volcanic forcing leads the PDO to a negative phase right after the eruptions (Fig. 10f). However, the Volcano&SolarStrong experiment leads to a stronger negative phase after volcanic eruptions compared to the Volcano experiment, especially for the Tambora eruption. This indicates that the solar forcing can enhance the PDO response from volcanic forcing, echoing the additional SST cooling transport to the northern extra-tropics found in the Volcano&SolarStrong experiment (section 4.1; Fig. 7). The AMV has clear negative signatures from all forcings as expected (Fig. 10g), but whether
this cooling response is truly an AMV signal is still under debate since the negative AMV may include the global volcanic cooling signature (Fang et al. 2021). No apparent signature is found in the AMOC (Fig. 10h).

## 5 The Arctic Amplification from Solar and Volcanic Forcings

In this section, we investigate the AA response to solar and volcanic forcing because the northern extratropical SAT cooling is strongly controlled by feedbacks associated with the AA. Figures 11a shows the strength of AA cooling over 1809-1826 by
the ratio between the polar surface temperature and the global surface temperature. The SolarStrong experiment has a stronger ensemble-mean AA cooling (3.2) compared to the Volcano (1.9) and Volcano&SolarStrong (2.1) experiments over 1809-1826. This difference is, however, not significant since a large spread is found in SolarStrong. This is because the global cooling is weak and unstable in individual ensemble members, which is also true in the Basic experiment. If we only consider the post-volcano period (1821-1826), the AA in the Volcano and Volcano&SolarStrong experiments increases and shows
similar values as in the SolarStrong experiment. This is because strong tropical cooling exists in the first 3-5 years corresponding to the direct volcanic forcing resulting in the lower AA right after the eruptions. That is, the strength of AA cooling caused by solar and volcano are comparable (though in different cooling magnitude) after the direct volcanic forcing diminish, while weak AA is found when the volcanic forcing exists.

To understand whether the AAs from solar and volcanic forcings are related to the same processes, we use the rapid adjustment
(Marshall et al. 2020) of each process over the period. This method obtains the radiative changes (W/m$^2$) in the system caused by each process (or feedback) by multiplying the radiative kernel with the difference to the reference model (see Method Section for details).  For instance, the albedo feedback is a positive feedback, meaning surface cooling results in more albedo, which increases the outgoing shortwave radiation at top of the atmosphere and thus reduce heat in the system  (negative values; see SW_albedo), while the temperature feedback is a negative feedback that increases the heat in the system (positive values;
see LW_ta and LW_temp2) by releasing less longwave radiation at top of the atmosphere. Figures 11c-f show the differences of rapid adjustments between the solar and/or volcano forcing experiment and the Basic experiment for the tropics and the Arctic averaged over two periods. The total increase of radiative difference in the Arctic over 1809-1826 is smaller than the



respective change in the tropics for all experiments, indicating a weaker recovery from the cooling in the Arctic. For SolarStrong, the albedo feedback is the most crucial process that compensates for the temperature feedback in the Arctic (Fig. 11c), while the compensation with longwave water vapor change is relatively small in the tropics (Fig. 11e). This shows that the sea-ice increase (and partly snow) contributes mainly to the AA in SolarStrong. For the Volcano experiment, besides the temperature feedback, the shortwave cloud radiative changes also contribute to warm the Arctic, which is suggested to be associated with the reduction of high-level clouds (Marshall et al. 2020). Although half of the changes are compensated by the longwave cloud feedback, the total cloud feedback is an additional strong negative feedback that helps the system return to its climatology. If we only consider the post-volcano period (1821-1826), we can see that the cloud feedback contributes little (Figs. 11d and f). Instead, the radiative changes from the albedo are maintained at ~70% of the period right after volcanic eruptions (Fig. 11d), which is the main feedback that keeps the Arctic colder than the tropics.

## 6 Discussion and Summary

This study quantifies how much each of the volcanic and solar forcing can separately contribute to the well-known early 19th century cooling and examines the combined climate responses from the two forcing agents with 20-member ensemble simulations from the MPI-ESM1.2 model. To our knowledge, this is the first study comparing the two solar reconstructions from the PMIP4 past1000 protocol as well as examining the interactions of solar and volcanic signals in the early 19th century with a large number of ensemble members. The solar forcing from the SATIRE solar reconstruction (Basic experiment) contributes less than 0.05 K/month SAT cooling on average, while the forcing with the PMOD solar reconstruction (SolarStrong experiment) contribute an additional 0.1 K/month SAT cooling. The 1809 unknown and 1815 Tambora eruptions contribute 0.82 K and 1.36 K SAT cooling at their peak. The cooling then rapidly returns to ~0.35K around the year 1820 and the temperature slowly approaches the climatological value. The cooling signatures are strongest in the northern extra-tropics due to the AA, except during the first few years after eruptions when the prescribed volcanic forcing directly cools the tropics. In addition, the SST cooling in the northern extra-tropics is characterized by a slow recovery after the direct volcano-induced cooling compared to the SAT and other regions. This slow recovery from the SST cooling is interpreted as a connection with the slow reduction of the Arctic sea-ice extent and accompanied by a smaller SST seasonality.

The additivity of the climate responses to the volcanic and solar forcing is studied by a sequence of simulations with combined and separated volcanic and solar forcing agents. We find that the surface climate responses are in general additive, but the responses on the regional scale can be non-additive. In the middle stratosphere, the reduced solar radiation and volcanic aerosol separately cause opposite responses of the polar vortex, but the response in combined simulations shows an additional strengthening of the polar vortex. This indicates that the solar forcing may enhance or have no impact when imposed on the volcano-induced polar vortex change. It has to be noticed that ozone is not included interactively and no QBO is simulated in this version of MPI-ESM. As both ozone (Shindell et al., 1999; Oehrlein et al., 2020) and QBO (Holten and Tan 1980;



Garfinkel et al. 2012; Labe et al. 2019) can impact the polar vortex, these processes may lead to more complex interactions

between solar and volcanic forcing, which is not considered in our study.

Model simulations generally show an El Niño signature right after tropical volcanic eruptions (Khodri et al. 2017); however, in our simulations the El Niño tendency is found to be weaker when including the solar forcing (though not significantly). This may be related to the La Niña tendency found during solar minimum (Lin et al. 2021) but no such tendency is found in our solar forcing-only simulations. That is, the solar irradiance may contribute to the tendency of ENSO events after eruptions and

give a hint on why no consistent ENSO responses from volcanoes have been found over the past centuries (Dee et al., 2020). Furthermore, when imposing reduced solar radiation in the volcano experiment, the additional tropical cooling is simulated right after the eruptions and then further propagates to the northern extra-tropics through ocean circulation and reduces the near-surface air temperature along the propagation. This indicates that solar forcing may have an additional impact on regional temperature change caused by volcanic eruptions, which further enhances the difficulty when combining or interpreting the

proxy reconstruction in different regions. Figure 12 shows northern extratropical summer land surface air temperature with multiple proxy reconstructions. There are large differences between reconstructions. Previous studies have stressed the importance of solar forcing in post-volcanic cooling (Anet et al., 2014), even though the early 19th century cooling is dominated by the eruptions (Wagner and Zorita, 2005). However, as shown in Figure 12, the large spread of the reconstructions cannot confirm the necessity to include solar forcing even when solar forcing can contribute to cooling in the post-volcano

period.

In addition, the AA response to volcano and solar forcing is investigated in this study, which is one of the limited studies about the cooling contribution of AA. We find that the albedo feedback is the main positive feedback (cooling causing further cooling) that slows the Arctic temperature from returning to climatology after volcanic eruptions. Since the sea-ice extent is not directly depending on the air temperature and is related to the interaction with SST, the sea-ice albedo feedback remains

at roughly 70% in the post-volcano period, indicating the importance of Arctic sea ice for the post-volcanic cooling.

This study shows that the comparably small solar forcing cannot be ignored for understanding the regional climate of the early 19th century, even though it has small global impacts compared to the two large volcanoes. This indicates that other small disturbances may also contribute to regional climate anomalies after strong tropical volcanic eruptions. For instance, small eruptions (or the latitudinal difference of eruptions), which are not included in the EVA volcano forcing from the PMIP4

past1000 protocol, may also play a role for regional changes.

## 7 Conclusion

This study is the first quantification of both solar and volcanic cooling in the early 19[th] century with large (20) ensembles and with two solar reconstructions. Besides the additive responses in general, regional impacts (such as polar vortex and surface temperature propagations) can be nonadditive when together or separately imposing the solar and volcanic forcing.





Furthermore, AA cooling, which is rarely discussed, is found in both solar and volcanic impacts, revealing the importance of the albedo feedback (or sea-ice change) in controlling the post-volcano cooling.

**Code Availability**

The Python code for generating the figures can be accessed at Zenodo with DOI: 10.5281/zenodo.6567188.

**Data Availability**

The processed data of the variables for each experiment can be accessed at Zenodo with DOI: 10.5281/zenodo.6567188.

**Author contributions**

SWF performed the model runs and the statistical analyses and wrote the paper. CT, JJ and HS designed the study. CT, JJ, KK, and HS contributed to writing the paper. All authors contributed to discussion and finalization of the article.

**Competing interests**

The authors have no competing interests.

**Acknowledgments**

We thank Andrea Schneidereit who gave valuable comments on an earlier version of this paper. The research is funded by the German Federal Ministry of Education and Research (BMBF), research programme "ROMIC-II, ISOVIC" (FKZ: 400  01LG1909B). C.T. is funded by the DFG research unit FOR 2820: Revisiting The Volcanic Impact on Atmosphere and Climate-Preparations for the Next Big Volcanic Eruption (VolImpact, grant number:398006378). K.K. acknowledges the contribution to this study by the Research Council Norway TOPPFORSK project VIKINGS (grant number #275191), which developed during her scientific guest stay at MPI-M financed by the Faculty of Mathematics and Natural Sciences of the University of Oslo. The computations, analysis and model data storage were mainly performed on the computer of the 405  Deutsches Klima Rechenzentrum (DKRZ) using resources granted by its Scientific Steering Committee (WLA) under project ID bb1171. We acknowledge the World Climate Research Programme's Working Group on Coupled Modeling, which is responsible for PMIP. The analyses were performed using Python.



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





**Figure 1: (a) Global mean sea surface temperature anomalies (K) of past2k simulation (black) and the perturbed 19 ensemble menbers. (b) Global mean incoming solar radiation anomalies at top-of-atmosphere (W m$^{-2}$) for the solar forcing of SATIRE-14C (Basic experiment) and PMOD-14C (SolarStrong experiment). (c) Global mean outgoing solar radiation at top-of-atmosphere (W m$^{-2}$) for experiments with (Volcano experiment) and without volcanic forcing (Basic experiment).**







**Figure 2: (a) Global mean SAT (K) anomalies of the Basic (blue) and SolarStrong (orange). (b) Tropical mean (30° S to 30° N), (c) northern extratropical mean (30° N to 90° N), and (d) southern extratropical mean (90° S to 30° S). (e)-(h) are for SST (K).The thick lines are the ensemble mean. The orange boxes below indicate periods during which the ensemble means of SolarStrong are significantly different (see Method Section for details) to the Basic experiment. The magenta vertical lines indicates the year of 1809 and 1815 Tambora eruptions, other gray vertical lines are the January for each year.**





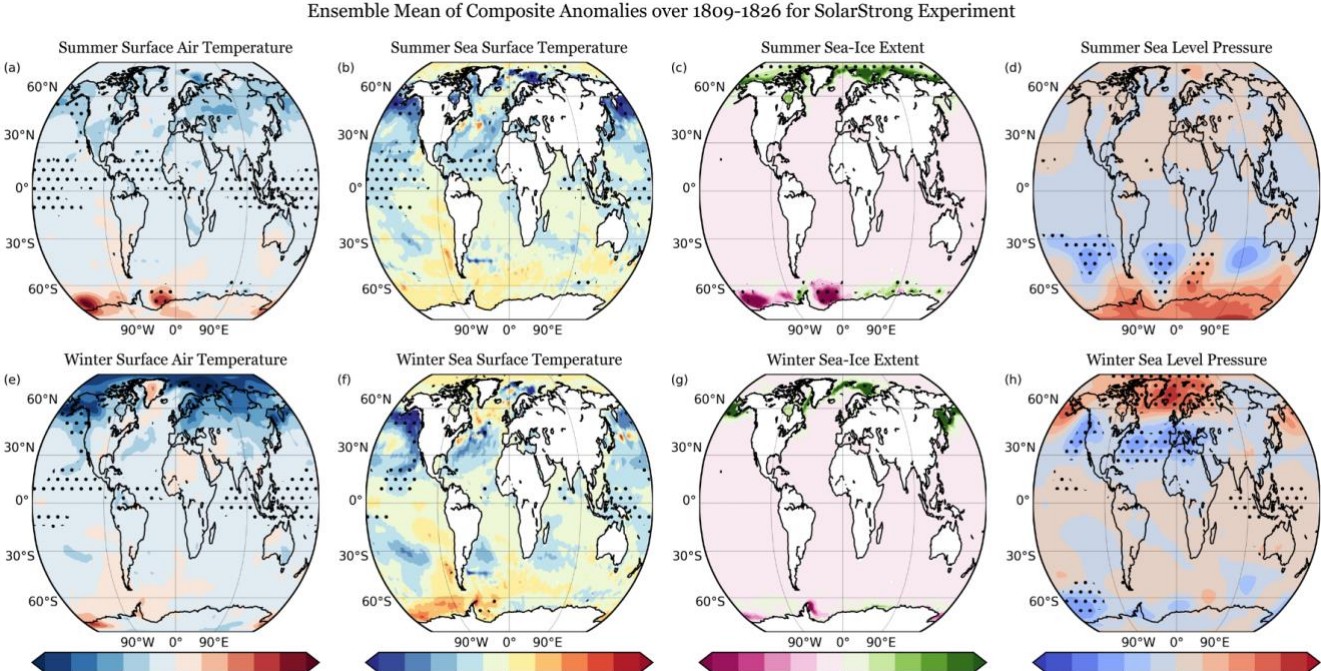

**Figure 3: Ensemble mean of composite of summer (a) SAT (K), (b) SST (K), (c) sea-ice are fraction (%), and (d) sea level pressure (Pa) anomalies over 1809-1826 for the SolarStrong experiment. (e)-(h) are for winter. The black dots represent the significant grid points (see Method Section for details).**








**Figure 4: As in Fig. 2 but for the Volcano experiment.**



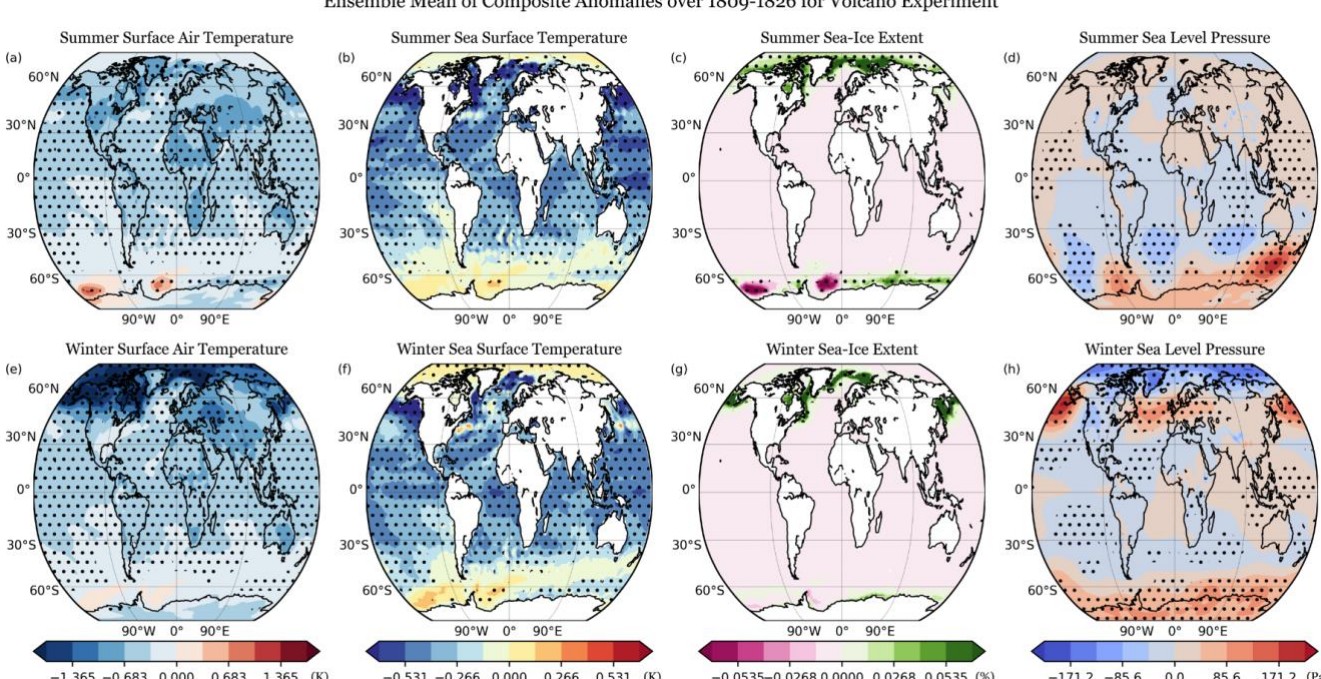

**Figure 5: As in Figure 3 but for the Volcano experiment.**







**Figure 6: (a) Ensemble mean global mean SAT (K) anomalies of the SolarStrong (orange), Volcano (green), and Volcano&SolarStrong (red) minus the Basic experiment. The purple dashed line is the sum of SolarStrong and Volcano (minus Basic) experiment. The gray vertical lines are the January for each year. (b) Tropical mean, (c) northern extratropical mean, and (d) southern extratropical mean. (e)-(h) are for SST (K).**





**Figure 7: (a) Hovmöller plot for zonal mean winter SAT anomaly (K) for the Basic experiment. (b) the SolarStrong, (c) SolarStrong minus Basic. The black dots illustrate the significance (see Method Section for details). (d) Volcano, (e) Volcano&SolarStrong, and (f) Volcano&SolarStrong minus Volcano. (g)-(l) are for SST (K).**



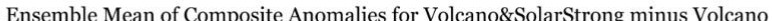

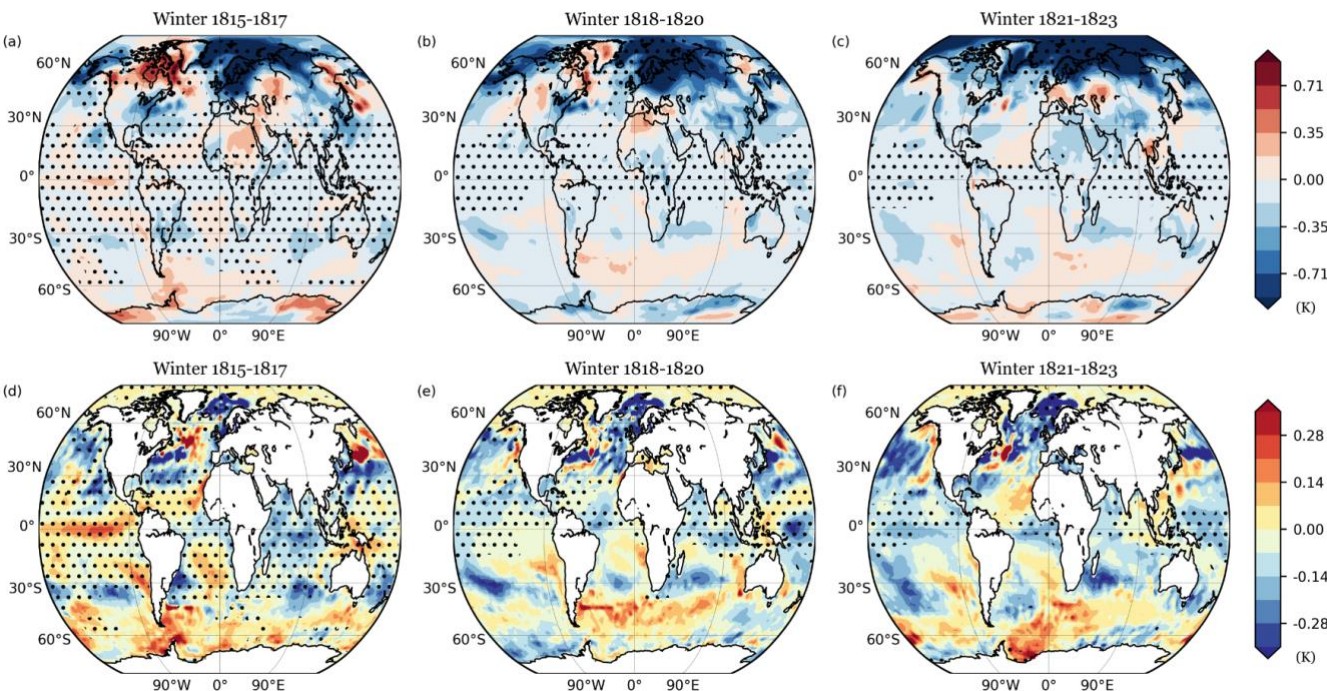

**Figure 8: (a) Composite of ensemble mean winter SAT (K) over 1815-1817 for the Volcano&SolarStrong experiment minus the Volcano experiment. And the block dots show the 95% significance between the Volcano&SolarStrong and Volcano experiment with t-test from 20 ensembles. (b) and (c) are for 1818-1820 and 1821-1823, respectively. (d)-(f) are for the SST (K).**



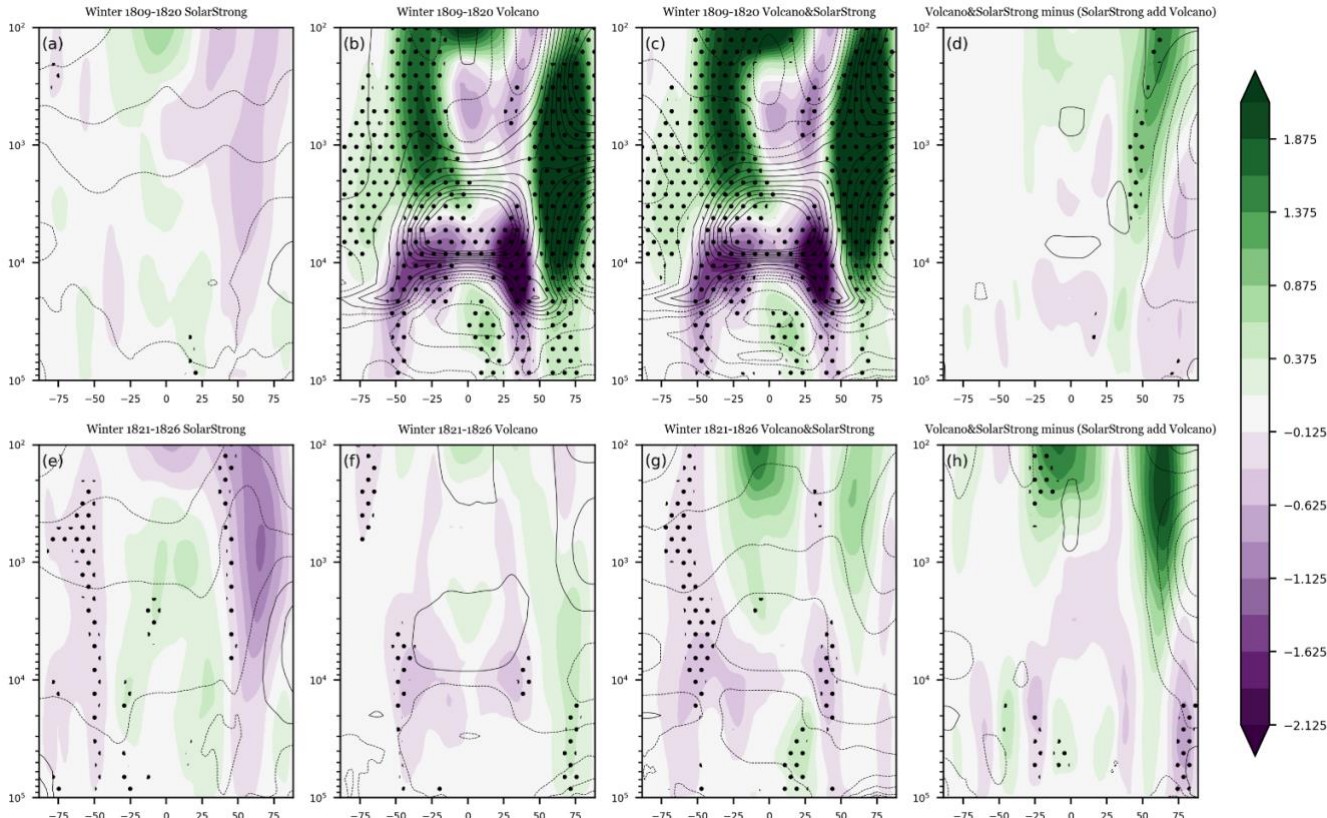

**Figure 9: (a) Ensemble mean of composite zonal mean temperature (K; contour with internal 0.25 K) and zonal winds (m/s; shading) anomalies over 1809-1820 winters for SolarStrong experiment. (b) Volcano, (c) Volcano&SolarStrong and (d) Volcano&SolarStrong minus (SolarStrong plus Volcano). The black dots in (a)-(c) show the 95% significance of zonal mean zonal winds compared to the Basic experiment, while in (d) shows the 95% significance between the Volcano&SolarStrong and (SolarStrong plus Volcano). (e)-(h) are for 1821-1826 winters.**







**Figure 10:** As in Figure 2, but for (a) the Niño3.4 index (K), (b) Relative Niño3.4 index (K), (c) Winter NAO index (Pa), (d) Winter AO index, (e) Summer SAM index (Pa), (f) PDO index, (g) AMV index (K), (h) AMOC index (kg s⁻¹). The Basic experiment is in blue, SolarStrong is in orange, Volcano is in green, and Volcano&SolarStrong is in red.







**Figure 11: (a) Ratio of Arctic SAT anomalies (67° N to 90° N) to global mean SAT anomalies over 1809-1826 for each ensemble in the Basic, SolarStrong, Volcano, and Volcano&SolarStrong experiment. The larger dot is the ensemble mean (ensemble mean Arctic SAT anomaly/ ensemble mean global SAT anomaly). (b) 1821-1826. (c) the difference between the Arctic (67° N to 90° N) radiative change (W m$^{-2}$) of each experiment to the radiative change of the Basic experiment over 1809-1826. (d) 1821-1826. (e) and (f) are for the Tropics (30° S to 30° N). The SW and LW mean shortwave and longwave radiation changes, respectively; and ta is air temperature difference; temp2 is surface air temperature; q is water vapour; albedo is albedo; c is cloud; total is the summation of the 7 radiation changes.**



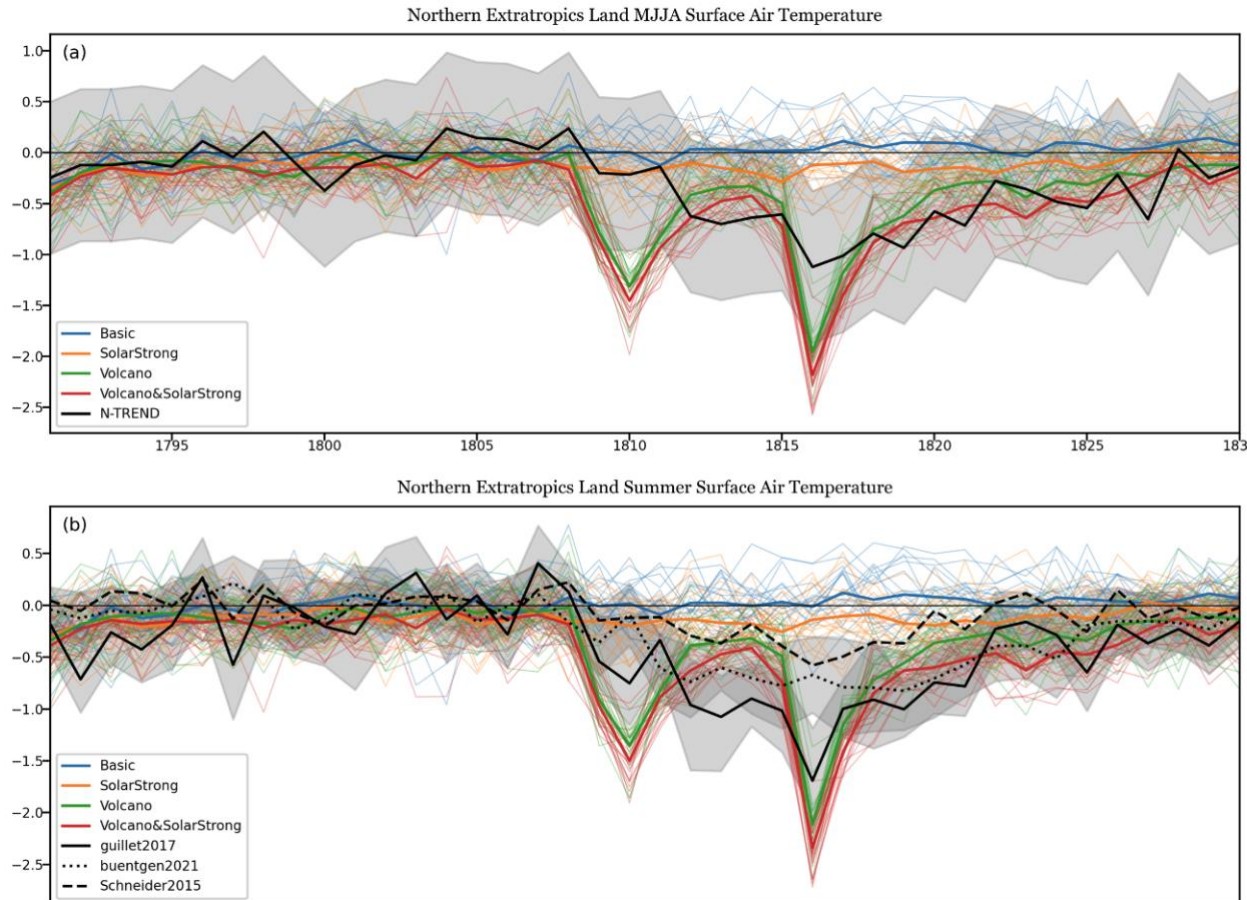

**Figure 12: (a) Northern extratropical (30° N to 90° N) May to August land surface air temperature anomalies Basic (blue), SolarStrong (orange), Volcano (green), and Volcano&SolarStrong (red) experiments and (b) for northern extratropical summer land surface air temperature. The black lines are for the anomalies from reconstruction projects and the gray shadings are the uncertainties. The reconstruction anomalies are offset with the difference from the ensemble mean of the Basic experiment over 1800-1808 for lining with the Basic experiment. In (a), the uncertainty of the N-TREND (Wilson et al., 2016) is provided in the dataset. In (b), the uncertainty of guillet2017 (solid; Guillet et al., 2017) is provided in the dataset and the uncertainty for the buentgen2021 is calculated by the one standard deviation of the 15 ensembles (R1 to R15; dotted; Büntgen et al, 2021). And Schneider2015 is in dashed (Schneider et al., 2015).**



| Experiment | Solar Forcing | Volcano Aerosol | Ensemble | Time |
|---|---|---|---|---|
| Basic (Climatology) | SATIRE-14C | None | 20 | 1791-1830 |
| SolarStrong | PMOD-14C | None | | |
| Volcano | SATIRE-14C | eVolv2k | | |
| Volcano&SolarStrong | PMOD-14C | eVolv2k | | |

**Table 1: Description of the four experiments used in this study.**

660