# Peer review of "On the Additivity of Climate Responses to the Volcanic and Solar Forcing in the Early 19th Century"

_EGUsphere, 2022_

## Referee Comment (RC2)

The manuscript discusses the additivity of the volcano and solar forcing (the latter using two different reconstructions of solar irradiance) in contributing to the cooling observed over the early 19th century. The paper includes interesting and detailed analysis, and the use of relatively large ensemble of MPI-ESM simulations (20 members per experiment) allows substantially narrow down the uncertainty arising from interannual variability. Some improvements in the manuscript would be needed, as detailed in the comments below, although in my opinion the manuscript should be published once these are addressed.

L11: suggested -> ', as suggested'

L12: by simulating -> that simulates

L13-14 by saying 'in general additive', do you mean global mean cooling is additive? (as opposed to regional changes, which are non additive). If so, this needs to be said explicitly, as 'in general' is too vague and confusing. This applies throughout the manuscript.

L15. I don't understand how do you obtain the units of cooling as K/month (here and in the main manuscript and summary) – feels a bit odd to express it like that in my opinion – either explain that in main text or use the maximum amplitude (as with volcanic eruptions)

L15 surface air cooling – near-surface air cooling

L17 cooling peak of -> cooling that peaks at

L17-19. Something is not fully correct with that sentence

L19-20. 'which is related to the reduction of seasonality and the increased Arctic sea-ice extent' -> 'which is related to the concurrent changes in Arctic sea-ice extent'. Not sure why do you note the reduction in seasonality here (which if I understood the manuscript correctly relates to the SST vs SAT cooling in the NH midlats)

L22 polar vortex -> stratospheric polar vortex

L23 opposite responses -> opposite-sign changes in stratospheric temperatures and zonal winds

L67. Distinct surface responses -> distinct regional and seasonal surface responses

L75-77. That's because of the dependence of the horizontal distribution of sulfate aerosols from volcanic eruption (i.e. how far north do the aerosols get), and the uncertainty in it, rather than because of some general uncertainty in AA response to volcanic forcing. You might want to clarify that.

L88. Atmospheric model top?

L105 'a larger variability' – I don't like the term as it makes it sound there is more natural variability in it. Perhaps change to something like 'larger long term changes in solar irradiance' (or similar)? That applies here and later on in the manuscript.

L102-115. One of the important things that needs to be added here is by how much does the UV radiation varies in the two reconstructions. Fig. 1 includes changes in total solar irradiance, which are most important for determining the bottom-up cooling as discussed in much of the manuscript, but since Section 4.2 discusses also the changes in the stratosphere (the 'top-down; mechanism) the

main thing that determines the response there is the UV change, and not TSI. Please add this to the manuscript.

Similarly, the paragraph talks about the prescribed ozone changes but what is unclear is whether the same ozone field is used for all 4 experiments (especially for the two solar reconstructions). The associated changes in stratospheric ozone have been found to cause ~50% of the stratospheric temperature response to the 11-year solar cycle (e.g. Gray et al., 2009), and so they will be crucial in determining the stratospheric response in Section 4.2

L118-L120. Please include the amount of SO2 injected for each eruption, as well as the altitude and latitude. Also, might be worth adding a figure, either to the main text or the supplement, that shows the latitude vs height concentrations of simulated sulfate, or at least latitudinal profile of AOD, to show the horizontal distribution of the volcanic forcing.

L140-149: might be worth explaining the rapid adjustment method further as it's still somewhat difficult to understand to a reader not familiar with the method.

L155-154. As mentioned above, unclear how do you arrive on the K/month unit. Please specify, or use a maximum amplitude to make it more comparable with the magnitude of volcanic forcing.

Fig.2 -> panel labels not included in the figure

L163: 'Maximum differences in solar irradiance and surface temperature do not coincide.' I don't fully agree - in the NH extratropics there is a good correlation between difference in TSI and temperature between the two reconstructions.

L166-167. 'This indicates that the solar forcing does not exhibit clear instantaneous response' -> add 'throughout the globe'?

L179 change to 'due to weaker westerly winds (Kwon and Joyce, 2013) and reduced freshwater flux'?

L182: 'accompanied with a smaller increase in the summer than in winter'. First, I don't think 'accompanied with' is the right word here. Second, is that right? I see as large (if not larger) increases in summer sea-ice as in winter sea-ice.

L185: 'weakening of the polar vortex due to the smaller meridional temperature gradient' -> 'weakening of the stratospheric polar vortex due to the reduction in the meridional temperature gradient'

L188. it's not a cause/effect, and rather the two things are consistent with each other.

L194-196. That would be a nice place to refer to the plot showing latitudinal distribution of sulfate aerosols or AOD, so that to illustrate where the volcanic forcing is applied.

L196: 'contribute' -> 'contribute comparatively more'

L196. 'and weaker (larger) seasonal variations are observed in the SST (SAT)' – make that a separate sentence.

L205-206. I would say it is also found in SH, it's just the initial perturbation (when aerosols are present) is so much smaller?

L214-215. 'That is, although different patterns are found in the sea-ice extent and surface pressure and the cooling magnitudes differ' – I'd say that the sea-ice extent response is similar between

volcano and solarstrong (just stronger, consistent with the stronger global mean cooling), it's just the high-lat SLP response that it different

L226-237: this paragraph needs to say upfront that the regional responses are non-additive. Also I'd argue that it's not only SST responses but also SAT responses that are non-additive, esp. in the tropics

Figure 9-> The contours are really difficult to read (both in terms of the magnitude and sign). Given the stratospheric temperature changes are critical to understand the zonal wind response, I think it would be very useful to add the changes in atmospheric temperatures (including statistical significance of these) as a separate plot (or extra 2 rows of panels in Fig. 9).

L.242: 'the reduced solar irradiance results in a weaker polar vortex' -> either use 'suggest' rather than 'result in' or note the response is not statistically significant.

L243: westerly anomalies -> easterly anomalies

L245: can cool -> can warm

L248: the warming in the tropical lower stratosphere -> add: from aerosol absorption

L253-257: please comment on how statistically significant is the difference between Volcano&Solar and (Volcano+Solar), and clarify how do you estimate this

L265-267: 'In addition, the signal is significant in the polar troposphere. This is related to a corresponding response in the same region in the Volcano experiment This indicates that the volcanic impact is more related to the surface, while the responses to the reduced solar radiation are less significant and may be more confined to the stratosphere.' -> First, what signal? Do you mean just the difference between Volcano+Solar and Volcano&Solar? Second: The way I see it, in earlier periods when volcanic forcing is strongest, the stratospheric response is overwhelmed by the warming from sulfate aerosols. But in the latter period, when aerosol forcing is smaller, there is more of an interplay of volcanic forcing (warming in the lower stratosphere driving a strengthening of the tropospheric jet) and solar forcing (cooling in the upper stratosphere driving a statistically insignificant easterly stratospheric jet response that can propagates down to the troposphere), which given non-linear nature of wave propagation in the stratosphere could (maybe) give you a somewhat different response when the two forcings are considered together.

L269. please comment also on whether the same ozone field is prescribed for both solar reconstructions, ( and if so, what effect this could have).

L272 not shown – perhaps useful to include into the supplement?

L272-273: 'as they are related to the surface ocean changes, which are also additive for solar and volcanic forcing' -> Do you mean *global mean* surface ocean responses (as opposed to regional ones that are non-additive)? If so state that clearly.

L282: 'has weak El Niño signatures for two years after both 1809 and 1815 eruptions,' -> first, 'only a weak'. Second, please clarify what do you mean by 'for two years' here, as to me it suggests that the El Nino should last for two years, although I don't think that's what the plot shows.

L283: 'may cause a smaller El Niño tendency' -> 'may reduce the tendency towards El Nino'

L267-268: 'No significant signal is simulated for the 1810 winter and not all the 1816 winter has a significant signal' -> First , start with 'However'. Second – what metric/experiment do you refer to

here? I see positive AO in both volcano and volcano&solar. Third - For NAO, it could be that the model variability doesn't really align with the regions chosen (from observations) as NAO 'centres of action' – have you checked whether under +ve AO the model shows significant +ve and -ve sea level pressure changes in the same exact regions chosen for NAO index?

L281-190: 'After the first year, when the northern extra-tropics also encounter stronger cooling, no significant impact is found for NAO and AO' -> as above, what experiments? I see clear positive AO.

L290-291: 'which agrees with the weak response in the composite of sea level pressure (Figs 3h and 5h)' -> to me these winter high-latitude SLP responses are actually quite strong/clear (and statistically significant)

L293-294: 'which is consistent with the mix of increasing and decreasing Antarctic sea-ice related to the surface pressure and temperature changes in the same regions' -> 'consistent with the mix of decreasing and increasing temperature and sea-ice in the same region.' SLP is a signature of SAM changes and that snows a tendency towards -ve SAM. In fact, SAM index for volcano&solar and volcano(post Tambora) shows a clear tendency toward -ve SAM, and this agrees with the SLP response in 5h. Please comment on that.

Also, does 'winter' in Fig 3/5 refer to DJF, and 'summer SAM' in Fig. 10 also to DJF? (as austral summer?). that needs to be clearer I think

L296: stronger -> stronger and longer lasting

L301: 'No apparent signature is found in the AMOC (Fig. 10h).' -> In most experiments yes, but the SolarStrong shows a pretty clear weakening of AMOC.

L303 'because' -> this is not the right word here I think

L304-L305: 'by the ratio' -> 'as approximated by the ratio'

L305-308: I think one of the main message here is that there is large variability in AA for basic and solar, whilst the variability becomes (surprisingly) very small for volcanic and volcanic+solar.

L309: 'the AA in the Volcano and Volcano&SolarStrong experiments increases' -> add 'and so does the intra-ensemble spread'

L311-313: 'That is, the strength of AA cooling caused by solar and volcano are comparable (though in different cooling magnitude) after the direct volcanic forcing diminish, while weak AA is found when the volcanic forcing exists.' -> That is, the strength of AA caused by solar and volcano is comparable (despite differences in absolute global mean cooling) after the direct volcanic forcing diminishes, while only weak AA is found when the volcanic forcing exists.

L320-332: I believe this section needs more explanation of the different feedback mechanisms to help readers less familiar with the literature

L343: AA -> I would avoid using acronyms in the summary

L344-346: the way I see these plots, it is both SST and SAT that are characterised by a slow recovery, esp, in the NH extratropics..

L347: sequence -> set

L348: surface climate responses -> global mean/large scale surface climate responses

L350: 'cause opposite responses of the polar vortex' -> 'cause opposite sign changes in stratospheric temperatures and thus the NH polar vortex'

L350-1: 'shows an additional strengthening of the polar vortex' -> 'shows a strengthening of the polar vortex of a similar or somewhat stronger magnitude to that in a volcanic simulation alone'

L351: This indicates that the solar forcing may enhance or have no impact when imposed on the volcano-induced polar vortex change-> This suggests that the solar forcing may have little impact on, or even enhance, the volcano-induced strengthening of the polar vortex

352-353: same prescribed ozone in all experiments?

L365+: I would argue that Figure 12 should be moved to the beginning of the manuscript to give the reader some context

L356-L360: Swap the order of these sentences: sentence 1, sentence 3, sentence 2.

L371: In addition -> Finally/Lastly?

L372: the cooling contribution of AA -> AA in the context of the global cooling? cooling contribution sounds a bit odd.

L383: additive responses in general -> additive responses in global mean/large scale

385: AA cooling ->> AA associated with the global cooling

Figure 1: Meaning of the vertical magenta lines?

---

## Author Comment (AC1)

**Shih-Wei Fang,** *Postdoctoral Scientist*
Bundesstraße 53, 20146 Hamburg, Germany
Tel.: +49 - (0)40 - 41173 - 239
shih-wie.fang@mpimet.mpg.de

October 15th, 2022

**The Editorial Board**
*Earth System Dynamics*

Dear Editors,

We would like to thank you for finding two reviewers whose constructive comments have helped us to improve the manuscript.

A large part of the reviewers' comments is related to clarifications of our results and model configuration. We have made revisions to the manuscript providing more detailed information to address the comments. For example, we have added a figure in Figure 1d to illustrate the near-UV change that is important for the top-down mechanism and changed Figures 1b and 1c from forcing responses to forcing variables to illustrate the real forcing used in the simulation. Also, the 20 century reanalysis data is included in Figure 12c as reviewer 1 suggested.

Other comments from both reviewers have also been addressed in the revised manuscript. None of these revisions affect the conclusions reported in the original manuscript in any substantial way.

Our point-by-point replies to the reviewers are attached below. The modifications made to address reviewers' comments are highlighted in red in the revised manuscript.

We hope you find the revised manuscript acceptable for publication in *Earth System Dynamics*.

Sincerely Yours,

Shih-Wei Fang and Co-authors

**Point-by-Point Replies to Reviewer #1**

We thank the reviewer for the very helpful comments, which broaden the scope of the study. Here are our point-to-point replies:

Summary

The authors analyse whether the volcanic and solar forcing response of the climate system is additive during the Dalton Minimum or not. Using ensemble simulations with a state of the art Earth System model, they find that the global climate responses are additive, but regionally some non-linear effect are present.

General comment

The authors present a well-structured study on a relevant topic, i.e., how natural forcing agents act separately and together on the climate system. The study is overall well written but the resolution of the figures need to be increased to at least 300 dpi. Overall, the study presents some new and interesting finding which desire publication in Climate of the Past, though I recommend some minor to major revision. One thing, which would make the study more convincing, is the inclusion of the 20CR reanalysis data which now spans back to 1806.

Reply: The reviewer has suggested the 20CR reanalysis data, which is now included in Figure 12c. We also raise the resolution of the figures as the reviewer suggested.

Specific comments

L13/14: Here the authors state that IN GENERAL the responses are additive. This is in contradiction to the statement later (L26-28) that regionally this is not the case. I suggest to write here that the superposition is only found for global mean considerations.
We have revised the 'in general' to 'in global mean/large scale' for a more accurate description, as well as other sentences we used the term 'in general'.
L22: The author state that the polar vortex strengthens when both forcing agents are present but weakens for each separate forcing agent. I did not find any explanation here or in the manuscript for this. The authors need to develop a mechanism/concept on why this is happening – just describing is not enough.
We have revised the sentences in Lines 271-274 to discuss the possible reasons: "This entails that the reduced solar radiation may contribute to a stronger polar vortex in the middle stratosphere (although we can identify only a few significant grid points) or be overwhelmed by the volcanic eruptions when strong volcanic eruptions occur. Further detailed studies will

need much larger ensemble members due to the strong internal variability in the Arctic climates (Liang et al. 2020)."

Liang, Y.-C., Kwon, Y.-O., Frankignoul, C., Danabasoglu, G., Yeager, S., Cherchi, A., Gao, Y., Gastineau, G., Ghosh, R., Matei, D., Mecking, J. V., Peano, D., Suo, L., and Tian, T.: Quantification of the Arctic Sea Ice-Driven Atmospheric Circulation Variability in Coordinated Large Ensemble Simulations, Geophysical Research Letters, 47, e2019GL085397, https://doi.org/10.1029/2019GL085397, 2020.

L36: change to "… (Timmreck et al., 2021) and the 1815   …"
Revised
L36-45: I think these studies are key for the discussion and should be used to really discuss the new findings of the manuscript in context to existing literature, so please use them in section 6.
We have revised the discussion and included a few papers along with the descriptions of reconstructions and reanalysis data in Lines 392-397.
L65-78: There is an interesting proposed by a colleague of mine which perfectly fits to the longer lasting effects after an external forcing event. In Lehner et al. (2013) they propose a sea ice-ocean-atmosphere feedback which can establish after a cooling in the Nordic Seas (induced by e.g. a volcanic eruption). So I think it is worth to mention it, to check whether this mechanism is also relevant in yoir study and to use it in the discussion part (Section 6).

Lehner et al., Amplified inception of European Little Ice Age by sea ice-ocean-atmosphere feedbacks. J. Climate, 26, 7586-7602, 2013.
We thank the reviewer for mentioning this paper. It provides an additional explanation of why the northern extratropical SST can remain cold for a long period. We have included the paper in Lines 217-219, where we discuss the SST responses to volcanic eruptions.

L97: change to "… 40 years. The 20 ensemble members are generated…"
Revised
L98: The authors need to explain how they perturb the atmosphere.
Added. 'with slight changes (0.99990 to 1.00009) in the stratospheric horizontal diffusion during 1771' in Lines 99-101.
L110: superscript 14 for 14C.
Revised
L125: Line break before "Several".
Revised
L139: Which significance level is used 5% 1% ????
5%, revised

L140-148: The description of the rapid adjustment remains unclear.

We have added sentences to further explain the rapid adjustment at Lines 154-156 and 339-340.

L148: we are considering -> we consider

Revised

L154-155: The authors give a rate her, i.e. k/month but is unclear over which period this rate is estimated.

We have changed the descriptions to 'a monthly average of 0.03 K global SST cooling by comparing before and after 1815 (< 0.05 K for SAT) in the MPI-ESM1.2-LR'. Other places using the rate are also changed accordingly.

L171: Winter -> winter

Revised

L171: I see that the temperature reduction is strongest in the Arctic in winter, but it is not significant, so why is that and why do the authors discuss not sign. results here. The only sign. response is the tropical temperatures showing a weak cooling.

We have added a sentence for the insignificance in the Arctic region at Lines 180-181: 'The large temperature change is not significant due to the large variabilities in the Arctic.' The tropical SAT change is mentioned with the SST at Lines 181-183.

L171-184: The entire discussion is strange. On the one hand the authors tend to focus on changes which are not significant and also ignore some strange behaviour, e.g., the Arctic sea ice in summer is strongly extended (significant) but the surface temperature shows almost no change, how can that be? Similar the AO reacts in winter but no imprint on the other variables. Figure 3 is not easy to compared with Fig 5 as the authors do not use the same colour scales.

We have revised a few sentences in 180-183 to clarify the main points.

For the summer temperature related to the Arctic sea ice increase, the significant results are mostly found in the polar region, where the temperature also cools at the same regions. For the AO responses, the limited imprint of the AO responses is due to the non-significant and weak amplitude of the AO index (Fig. 10d), even though the sea level pressure shows the tendency of AO, which we already mentioned in Lines 260-262.

L186: Yes, but the meridional temperature gradient at the surface is enhanced, which leads to higher lower tropospheric baroclinicity. I think the authors need to be clear that their model tends to show the top down process more pronounced compared to the surface processes.

The SAM does not have strong responses to solar forcing, which entails a tight interaction between the changes in sea ice, temperature, and surface wind. As a result, we are not showing the top-down process is more pronounced compared to the surface process, instead, our model results have shown a stronger bottom-up response as mentioned in Lines 231-232.

L191: Aerosols?

Revised

L195-199: I think here the mechanism described by Lehner et al. (2013) would be interesting

to be assessed.

We have decided to discuss Lehner et al. (2013) in Lines 217-219 along with the discussion of Carton et al. (2015).

L214-217: So if the authors discuss it in anyway in section 4.2 it is not necessary to give a presentation of the results here, so I suggest to remove this part or move and merge it with section 4.2.

The sentences are not only about the AO but also summarize the section. So we decided to keep the sentences.

L252: Line break after "forcing.".

Revised

L252-257: Only a few points are stat. significant at the 5% level in Fig. 9d. One would expect that 5% of the grid points are significantly different assuming independence so the authors cannot interpret the results of this panel to be different to pure noise.

Added, '(with a few significances found)' to indicate the possibility of over-interpretation.

L252-270 Interpretation of Fig.9: I think the authors tend to over interpret the significant changes, as only 50 % of the panels show clear signals and the rest can be seen as white noise (or no change). I suggest that the author revise this paragraph and concentrate on the significant changes in panels b, c, e, and g of Fig. 9.

We agreed with the reviewer about the limited significance when calculating the difference. We have revised a few sentences (Lines 268-290) to notify the readers of this limited significance issue. However, as the additivity is one of our main discussions in the manuscript, we decided to keep them.

L300-301: What about the wind driven gyre circulations – are there signals to be found?

As shown in the AO and NAO indices, the wind responses are not stably impacting the gyre and several papers have shown the typical AMV index contains the global cooling pattern from volcanic forcing.

L333: Please change to "Summary and Discussion"

Revised

L347-349: This a much better formulation than the corresponding one in the abstract.

We have revised the sentence in the abstract to better convey our results.

L335-380: The discussion part needs to be substantially extended by the publications used in the introduction (see above) and Lehner et al. (2013).

We have included the Lehner et al. (2013) in Lines 217.

L361: "… experiment, an additional tropical cooling …"

Revised

L365-370: A discussion here with using the 20CR reanalysis data would be very helpful here. I clearly recommend that the authors should compare their results with this reanalysis as it goes back to 1806 and at least over Europe it is reasonable.

We have included the 20CR reanalysis in Figure 12c and added the according discussion sentences in Lines 392-393.

L371-372: The sentence is awkward, so please revise it.

Revised to:

"Lastly, the Arctic Amplification response to volcano- and solar-forced global cooling is investigated in this study. This colder Arctic compared to the globe is rarely discussed since the warming perspective is generally the focus, such as Arctic Amplification responses to global warming" in Lines 400-402.

L373-375: Here you can include the discussion of the Lehner process as they involve also sea ice changes in the GIN Sea.

We have included Lehner et al. (2013) in Line 405.

L381-386: Sorry to say but this is not a conclusion so please write a proper one.

We have revised the conclusion in Lines 413-426.

L409 and following: I only checked the first 2 publications and important information is missing. Here, it is the journal name. I have not checked the rest but now-a-days there are a lot of tools helping the authors to avoid errors in the reference lists, so please, please, use these tools.

We have followed the suggested procedure by EGU for generating citations with Zotero. We will work with the editorial helpers and see which part of the information is missing if the manuscript is accepted.

Fig 2: Caption: please change to "…   (e)-(h) are for SST (K) and the same regions. The thick …"

Revised

Fig 3, 5 and maybe 8: It looks like the authors changed the aspect ratio of the panels, which lead to a strange projection. Please use the original aspect ratio as the projection should be kept. Please also Say the significance level here. By the way, you have two options either you say "the 5% significance level" or "the 95% confidence interval".   Other combinations (like "95% significance level") makes no sense (in statistics).

Due to the large amplitude differences between responses to the solar and volcanic forcing, it is better to change the scale of the color bars. We have added the number of scale changes (two times) compared to other figures. And the descriptions about significance are changed to '5% significance level' in the manuscript.

Fig. 7: Only few of points are significant so what do we learn from this figure (and the text in the manuscript). I suggest to remove it and also adjust the text. I recommend that the authors should stick to stat. significant results.

As the solar responses are small, limited significant results are expected even with 20 ensembles. We have decided to better inform the readers about the insignificance and not remove them. And Figure 7 shows how the few significant results may illustrate the difficulty of simulating

regional climate changes.

Fig.9: change to 5% significance level. (also throughout the manuscript).

Revised

Fig 12 Here I suggest to include also 20CR reanalysis or add it as a separate panel. In the caption at the end: ", and Schneider2015 is dashed (Schneider et al., 2015)."

We have included the 20CR reanalysis with an additional figure (Fig. 12c) and have discussed it accordingly.

**Point-by-Point Replies to Reviewer #2**

We thank the reviewer for the precise and constructive comments. Here are our point-to-point replies:

The manuscript discusses the additivity of the volcano and solar forcing (the latter using two different reconstructions of solar irradiance) in contributing to the cooling observed over the early 19th century. The paper includes interesting and detailed analysis, and the use of relatively large ensemble of MPI-ESM simulations (20 members per experiment) allows substantially narrow down the uncertainty arising from interannual variability. Some improvements in the manuscript would be needed, as detailed in the comments below, although in my opinion the manuscript should be published once these are addressed.

L11: suggested -> ', as suggested'

Revised

L12: by simulating -> that simulates

Revised

L13-14 by saying 'in general additive', do you mean global mean cooling is additive? (as opposed to regional changes, which are non additive). If so, this needs to be said explicitly, as 'in general' is too vague and confusing. This applies throughout the manuscript.

We are replacing the 'in general additive' in the manuscript with "global mean/large scale" as suggested.

L15. I don't understand how do you obtain the units of cooling as K/month (here and in the main manuscript and summary) – feels a bit odd to express it like that in my opinion – either explain that in main text or use the maximum amplitude (as with volcanic eruptions)

We have clarified the confusion of the unit by adding the period we calculated the average. We used this unit since the cooling is not stable over time and we decided to average over the cooling period.

L15 surface air cooling – near-surface air cooling

Revised.

L17 cooling peak of -> cooling that peaks at

Revised.

L17-19. Something is not fully correct with that sentence

Changed to:

"After the Tambora eruption, the temperature in most regions increases toward climatology largely within 5 years, along with the reduction of volcanic forcing" in Lines 17-19.

L19-20. 'which is related to the reduction of seasonality and the increased Arctic sea-ice extent' -> 'which is related to the concurrent changes in Arctic sea-ice extent'. Not sure why do you note the reduction in seasonality here (which if I understood the manuscript correctly relates to the SST vs SAT cooling in the NH midlats)

Changed to:

"In the northern extratropical oceans, the temperature increases slowly at a constant rate until 1830, which is related to the reduction of seasonality and the concurrent changes in Arctic sea-ice extent" in Lines 19-20.

L22 polar vortex -> stratospheric polar vortex

Revised

L23 opposite responses -> opposite-sign changes in stratospheric temperatures and zonal winds

Revised

L67. Distinct surface responses -> distinct regional and seasonal surface responses

Revised

L75-77. That's because of the dependence of the horizontal distribution of sulfate aerosols from volcanic eruption (i.e. how far north do the aerosols get), and the uncertainty in it, rather than because of some general uncertainty in AA response to volcanic forcing. You might want to clarify that.

We agree with the reviewer about the possibility of latitudinal distribution. We decided, however, not to mention this possibility since Liu et al. (2018) did not show the latitudinal distribution of their volcanic forcing but only referred to Gao et al. (2008). As a result, we do not know how different is their latitudinal distribution compared to our simulations.

Liu, F., Zhao, T., Wang, B., Liu, J., and Luo, W.: Different Global Precipitation Responses to Solar, Volcanic, and Greenhouse Gas Forcings, 123, 4060–4072, https://doi.org/10.1029/2017JD027391, 2018.

Gao, C., Robock, A., & Ammann, C. (2008). Volcanic forcing of climate over the past 1500 years: An improved ice core-based index for climate models. *Journal of Geophysical Research: Atmospheres*, *113*(D23).

L88. Atmospheric model top?

The top of the atmospheric model is 0.01 hPa or 80 km on average. We have added the description in Lines 89-90.

L105 'a larger variability' – I don't like the term as it makes it sound there is more natural variability in it. Perhaps change to something like 'larger long term changes in solar irradiance' (or similar)? That applies here and later on in the manuscript.

Revised

L102-115. One of the important things that needs to be added here is by how much does the UV radiation varies in the two reconstructions. Fig. 1 includes changes in total solar irradiance, which are most important for determining the bottom-up cooling as discussed in much of the manuscript, but since Section 4.2 discusses also the changes in the stratosphere (the 'top-down; mechanism) the main thing that determines the response there is the UV change, and not TSI.

Please add this to the manuscript.

Added to Figure 1d

Similarly, the paragraph talks about the prescribed ozone changes but what is unclear is whether the same ozone field is used for all 4 experiments (especially for the two solar reconstructions). The associated changes in stratospheric ozone have been found to cause ~50% of the stratospheric temperature response to the 11-year solar cycle (e.g. Gray et al., 2009), and so they will be crucial in determining the stratospheric response in Section 4.2

We have calculated the zonal average difference between the PMOD and the SATIRE in 1815 when their solar difference is the largest. And the SATIRE ozone has a maximum of 6% larger than PMOD in the Antarctic stratosphere and other differences are mostly ranging between 1-2%.

We have added a sentence to include this information in Lines 116-117:

"For instance, the zonal average ozone concentration calculated from the PMOD is roughly 2% different compared to the SATIRE ozone in 1815."

L118-L120. Please include the amount of SO2 injected for each eruption, as well as the altitude and latitude. Also, might be worth adding a figure, either to the main text or the supplement, that shows the latitude vs height concentrations of simulated sulfate, or at least latitudinal profile of AOD, to show the horizontal distribution of the volcanic forcing.

The model takes directly the AOD changes without the amount of SO2 injection, we have clarified the sentence as follows in Lines 123-125.

"Following the PMIP4 past1000 protocol, the time-varying aerosol optical depth (AOD) for sulfur injections of volcanic forcing is calculated with the EVA (Easy Volcanic Aerosol) module (Toohey et al., 2016) using the eVolv2k dataset (Toohey and Sigl, 2017) as input."

We have changed Figure 1c to the time-vary AOD since the information is similar to the original Figure 1c of the outgoing solar radiation.

L140-149: might be worth explaining the rapid adjustment method further as it's still somewhat difficult to understand to a reader not familiar with the method.

We have added a sentence to better explain the rapid adjustment in Lines 154-156.

L155-154. As mentioned above, unclear how do you arrive on the K/month unit. Please specify, or use a maximum amplitude to make it more comparable with the magnitude of volcanic forcing.

We have added the period we calculated the averages in Lines 163. And we have changed our description to 'a monthly average of 0.03 K global SST cooling by comparing before and after 1815 (< 0.05 K for SAT).

Fig.2 -> panel labels not included in the figure

The panel labels are in the left-up corner of each sub-figure.

L163: 'Maximum differences in solar irradiance and surface temperature do not coincide.' I

don't fully agree - in the NH extratropics there is a good correlation between difference in TSI and temperature between the two reconstructions.

The NH extra-tropics does not have a significant result in 1815 when the solar difference peaks. We have modified the sentence in Lines 172-173 to the following: 'Maximum differences in solar irradiance and surface temperature do not coincide instantaneously.'.

L166-167. 'This indicates that the solar forcing does not exhibit clear instantaneous response' -> add 'throughout the globe'?

Added

L179 change to 'due to weaker westerly winds (Kwon and Joyce, 2013) and reduced freshwater flux'?

Revised

L182: 'accompanied with a smaller increase in the summer than in winter'. First, I don't think 'accompanied with' is the right word here. Second, is that right? I see as large (if not larger) increases in summer sea-ice as in winter sea-ice.

Removed 'accompanied'. The summer sea-ice increases less in the winter if we compute the northern extratropical averages.

L185: 'weakening of the polar vortex due to the smaller meridional temperature gradient' -> 'weakening of the stratospheric polar vortex due to the reduction in the meridional temperature gradient'

Revised

L188. it's not a cause/effect, and rather the two things are consistent with each other.

We have revised the sentence to avoid the cause/effect interpretation for the sentence.

L194-196. That would be a nice place to refer to the plot showing latitudinal distribution of sulfate aerosols or AOD, so that to illustrate where the volcanic forcing is applied.

We have changed Figure 1c to the latitude-time AOD figure.

L196: 'contribute' -> 'contribute comparatively more'

Revised

L196. 'and weaker (larger) seasonal variations are observed in the SST (SAT)' – make that a separate sentence.

Revised

L205-206. I would say it is also found in SH, it's just the initial perturbation (when aerosols are present) is so much smaller?

We agreed there are responses in the first few years after eruptions, and have revised the sentence as follows in Lines 219-220:

'The reduced SST seasonality with the colder summer and the steady retrieval of the cooling is found in the southern hemisphere only the first few years after eruptions, as the Antarctic sea ice shows both increases and decreases depending on the region.'

L214-215. 'That is, although different patterns are found in the sea-ice extent and surface

pressure and the cooling magnitudes differ' – I'd say that the sea-ice extent response is similar between volcano and solarstrong (just stronger, consistent with the stronger global mean cooling), it's just the high-lat SLP response that it different

We agreed the sea-ice extent has also similar patterns, even though they are more extended to the south for the volcano runs. We have revised the sentence accordingly in Lines 229-232:

'That is, although differences are found between the Volcano and SolarStrong experiment, such as more southward extension of the sea-ice extent, opposite surface pressure patterns, and the cooling magnitudes, the response patterns of surface temperature are similar. This similarity indicates that the top-down mechanism via the AO has limited impacts and the surface is more dominated by the bottom-up direct radiative cooling.'

L226-237: this paragraph needs to say upfront that the regional responses are non-additive. Also I'd argue that it's not only SST responses but also SAT responses that are non-additive, esp. in the tropics

We agreed the SAT responses are also non-additive outside of the polar region and limit the discussion in the polar region. And we have emphasized the regional responses.

Figure 9-> The contours are really difficult to read (both in terms of the magnitude and sign). Given the stratospheric temperature changes are critical to understand the zonal wind response, I think it would be very useful to add the changes in atmospheric temperatures (including statistical significance of these) as a separate plot (or extra 2 rows of panels in Fig. 9).

Extra panels are added in Figure 9.

L.242: 'the reduced solar irradiance results in a weaker polar vortex' -> either use 'suggest' rather than 'result in' or note the response is not statistically significant.

Revised

L243: westerly anomalies -> easterly anomalies

Revised

L245: can cool -> can warm

Revised

L248: the warming in the tropical lower stratosphere -> add: from aerosol absorption

Added

L253-257: please comment on how statistically significant is the difference between Volcano&Solar and (Volcano+Solar), and clarify how do you estimate this

We have revised the sentences to:

'This entails that the reduced solar radiation may contribute to a stronger polar vortex in the middle stratosphere (although we can identify only a few significant grid points) or be overwhelmed by the volcanic eruptions when strong volcanic eruptions occur. Further detailed studies will need much larger ensemble members due to the strong internal variability in the Arctic climates (Liang et al. 2020)' in Lines 271-274.

L265-267: 'In addition, the signal is significant in the polar troposphere. This is related to a

corresponding response in the same region in the Volcano experiment This indicates that the volcanic impact is more related to the surface, while the responses to the reduced solar radiation are less significant and may be more confined to the stratosphere.' -> First, what signal? Do you mean just the difference between Volcano+Solar and Volcano&Solar? Second: The way I see it, in earlier periods when volcanic forcing is strongest, the stratospheric response is overwhelmed by the warming from sulfate aerosols. But in the latter period, when aerosol forcing is smaller, there is more of an interplay of volcanic forcing (warming in the lower stratosphere driving a strengthening of the tropospheric jet) and solar forcing (cooling in the upper stratosphere driving a statistically insignificant easterly stratospheric jet response that can propagates down to the troposphere), which given non-linear nature of wave propagation in the stratosphere could (maybe) give you a somewhat different response when the two forcings are considered together.

1. Yes.

2. We agreed with the reviewer's comment that these sentences are not clear enough. We have revised the sentences and included what the reviewer mentioned above in Lines 282-286.

L269. please comment also on whether the same ozone field is prescribed for both solar reconstructions, ( and if so, what effect this could have).

We have added the following sentence to Lines 287-290: "In our simulations, there is on average 2% difference in the stratospheric ozone between the calculations from SATIRE and PMOD reconstructions, which is within the observed variability (Fioletov et al., 2002) and should not have statistically detectable temperature changes."

Fioletov, V. E., Bodeker, G. E., Miller, A. J., McPeters, R. D., and Stolarski, R.: Global and zonal total ozone variations estimated from ground-based and satellite measurements: 1964–2000, Journal of Geophysical Research: Atmospheres, 107, ACH 21-1-ACH 21-14, https://doi.org/10.1029/2001JD001350, 2002..

L272 not shown – perhaps useful to include into the supplement?

We are not intended to provide any supplement for this paper.

L272-273: 'as they are related to the surface ocean changes, which are also additive for solar and volcanic forcing' -> Do you mean global mean surface ocean responses (as opposed to regional ones that are non-additive)? If so state that clearly.

We have revised it for clarity in Lines 292-293.

L282: 'has weak El Niño signatures for two years after both 1809 and 1815 eruptions,' -> first, 'only a weak'. Second, please clarify what do you mean by 'for two years' here, as to me it suggests that the El Nino should last for two years, although I don't think that's what the plot shows.

The small two spikes are what we are mentioning. We have revised the text to summarize the NAO and AO impacts in Lines 307-311.

L283: 'may cause a smaller El Niño tendency' -> 'may reduce the tendency towards El Nino'

Revised

L267-268: 'No significant signal is simulated for the 1810 winter and not all the 1816 winter has a significant signal' -> First, start with 'However'. Second – what metric/experiment do you refer to here? I see positive AO in both volcano and volcano&solar. Third - For NAO, it could be that the model variability doesn't really align with the regions chosen (from observations) as NAO 'centres of action' – have you checked whether under +ve AO the model shows significant +ve and -ve sea level pressure changes in the same exact regions chosen for NAO index?

1. Added.

2. We refer to the classical NAO and AO indices. We have also tested other NAO and AO indices but the responses are similar.

3. The NAO index calculated from the latitudinal difference of the geopotential heights at 500 hPa is also tested and the response is similar.

L281-190: 'After the first year, when the northern extra-tropics also encounter stronger cooling, no significant impact is found for NAO and AO' -> as above, what experiments? I see clear positive AO.

We are describing the responses after the peaks of positive NAO and AO. We have revised the sentence to summarize the responses of NAO and AO in Lines 307-311.

L290-291: 'which agrees with the weak response in the composite of sea level pressure (Figs 3h and 5h)' -> to me these winter high-latitude SLP responses are actually quite strong/clear (and statistically significant)

We have stressed the longer period of the composite to describe the responses in Line 310.

L293-294: 'which is consistent with the mix of increasing and decreasing Antarctic sea-ice related to the surface pressure and temperature changes in the same regions' -> 'consistent with the mix of decreasing and increasing temperature and sea-ice in the same region.' SLP is a signature of SAM changes and that snows a tendency towards -ve SAM. In fact, SAM index for volcano&solar and volcano(post Tambora) shows a clear tendency toward -ve SAM, and this agrees with the SLP response in 5h. Please comment on that.

Also, does 'winter' in Fig 3/5 refer to DJF, and 'summer SAM' in Fig. 10 also to DJF? (as austral summer?). that needs to be clearer I think

We added there is little response with the SAM and added discussions accordingly in Lines 311-313.

L296: stronger -> stronger and longer lasting

Added

L301: 'No apparent signature is found in the AMOC (Fig. 10h).' -> In most experiments yes, but the SolarStrong shows a pretty clear weakening of AMOC.

Agreed. We have added 'except a weak reduction tendency is found in the SolarStrong experiment' in Lines 321-322.

L303 'because' -> this is not the right word here I think

We agreed with the reviewer's comment and deleted the part for 'because'.

L304-L305: 'by the ratio' -> 'as approximated by the ratio'

Added.

L305-308: I think one of the main message here is that there is large variability in AA for basic and solar, whilst the variability becomes (surprisingly) very small for volcanic and volcanic+solar.

Agreed.

L309: 'the AA in the Volcano and Volcano&SolarStrong experiments increases' -> add 'and so does the intra-ensemble spread'

Added.

L311-313: 'That is, the strength of AA cooling caused by solar and volcano are comparable (though in different cooling magnitude) after the direct volcanic forcing diminish, while weak AA is found when the volcanic forcing exists.' -> That is, the strength of AA caused by solar and volcano is comparable (despite differences in absolute global mean cooling) after the direct volcanic forcing diminishes, while only weak AA is found when the volcanic forcing exists.

Revised

L320-332: I believe this section needs more explanation of the different feedback mechanisms to help readers less familiar with the literature

I have added a sentence to further explain the rapid adjustment in Lines 339-340.

L343: AA -> I would avoid using acronyms in the summary

Revised

L344-346: the way I see these plots, it is both SST and SAT that are characterised by a slow recovery, esp, in the NH extratropics..

I have revised the sentence to better express the larger slow recovery in Line 363.

L347: sequence -> set

Revised

L348: surface climate responses -> global mean/large scale surface climate responses

Revised

L350: 'cause opposite responses of the polar vortex' -> 'cause opposite sign changes in stratospheric temperatures and thus the NH polar vortex'

Revised

L350-1: 'shows an additional strengthening of the polar vortex' -> 'shows a strengthening of the polar vortex of a similar or somewhat stronger magnitude to that in a volcanic simulation alone'

Revised

L351: This indicates that the solar forcing may enhance or have no impact when imposed on the volcano-induced polar vortex change-> This suggests that the solar forcing may have little

impact on, or even enhance, the volcano-induced strengthening of the polar vortex

Revised

352-353: same prescribed ozone in all experiments?

The prescribed ozone is based on which solar reconstruction it uses. We have added information about how different they are in Lines 116-117.

L365+: I would argue that Figure 12 should be moved to the beginning of the manuscript to give the reader some context

We remain Figure 12 in the discussion since comparing with the observation is not the main topic for this manuscript.

L356-L360: Swap the order of these sentences: sentence 1, sentence 3, sentence 2.

We think the original order has a clear message.

L371: In addition -> Finally/Lastly?

Revised to 'Lastly'

L372: the cooling contribution of AA -> AA in the context of the global cooling? cooling contribution sounds a bit odd.

Revised to 'the Arctic Amplification response to volcano- and solar-forced global cooling' in Line 400.

L383: additive responses in general -> additive responses in global mean/large scale

Revised

L385: AA cooling ->> AA associated with the global cooling

Revised

Figure 1: Meaning of the vertical magenta lines?

Added

---

## Author Response (AR2)

**Point-by-Point Replies to the Editor**

We thank the editor for taking care of the manuscript and providing useful comments. Here are our point-to-point replies:

The authors have done a nice job with the revisions. I only have two comments:

1) It looks like Lehner et al. never made it into your references list. Check this and others.

Reply: We have checked the manuscript and Lehner et al. is in the reference list in Lines 526-527, hiding between the two new references.

2) Your response to Reviewer #2, Lines 75-77 is a little dissatisfactory. It's okay if you don't know how different the latitudinal distribution is. It's better to be honest and state the possibility rather than appear as though you're hiding it.

We agreed with the editor and revised the sentence as "Liu et al. (2018) state that no volcano-induced AA is found due to the large temperature changes in the tropics, which may also be related to the latitudinal distribution of volcanic forcing, while others have found that the northern extra-tropics encounter a larger cooling a few years after eruptions (Zanchettin et al., 2019)."